# Consequences of Circadian Disruption in Shift Workers on Chrononutrition and their Psychosocial Well-Being

**DOI:** 10.3390/ijerph17062043

**Published:** 2020-03-19

**Authors:** Nor Amira Syahira Mohd Azmi, Norsham Juliana, Nur Islami Mohd Fahmi Teng, Sahar Azmani, Srijit Das, Nadia Effendy

**Affiliations:** 1Faculty of Medicine and Health Sciences, Universiti Sains Islam Malaysia, Pandan Indah 55100, Malaysia; amirasyahira188@gmail.com (N.A.S.M.A.); drazmanisahar@usim.edu.my (S.A.); nadia@usim.edu.my (N.E.); 2Faculty of Health Sciences, Universiti Teknologi MARA, Bandar Puncak Alam 42300, Malaysia; nurislami@uitm.edu.my; 3Department of Anatomy, Universiti Kebangsaan Malaysia Medical Centre, Cheras 56000, Malaysia; srijit@ukm.edu.my

**Keywords:** workers, shift, biological clock, circadian rhythm, diet, psychosocial

## Abstract

The workers and employees in various institutions are subjected to different shifts and work schedules. The employees work not only at daytime but also during odd hours at night. The biological clock of an individual is often altered during night shifts. This affects the psychosocial well-being and circadian nutritional intake of the worker. Disturbance in circadian rhythm results in the development of metabolic disorders such as hypertension, dyslipidemia, dysglycemia, and abdominal obesity. In the present review, we discuss the nature of shift work, sleep/wake cycle of an individual, chrononutrition, dietary habits, and meal changes with regard to timing and frequency, related to shift work. We also discuss the relationship between nutritional intake and psychosocial well-being among shift workers. The review may be beneficial for prevention of metabolic disorders and maintaining sound psychological condition in shift workers.

## 1. Introduction

Throughout the world, twenty-four-hour services are a thriving part of the community. In order to meet the continuous demand of the urban world, crucial services are established by several industries and business establishments, which operate on 24 h basis. It is a necessity for many industries, including healthcare, transport, mining, and aviation, to have staff available for 24 h a day. These industries demand workers to perform significant tasks over a 24 h period [1]. Hence, it causes workers to routinely work on the basis of shift schedules [2]. Working in shifts consequently leads to an alteration in the biological clock. This affects the psychosocial well-being and circadian nutritional intake. Chrononutrition is a new area of study that emphasizes the interaction between nutritional intake and time of eating [3].

The worrying fact is that disruption of circadian rhythm often leads to increased risk of metabolic syndrome. Health problems such as hypertension, dyslipidemia, dysglycemia, and abdominal obesity occurring secondary to insulin resistance are commonly seen [4]. One of the reasons for the health consequences of shift work may be that food becomes a 24 h activity in a 24 h culture [5]. While those on a more conventional daytime work schedule are most likely to eat three meals every 24 h, with food consumed during daytime [6,7], shift work usually contributes to altered eating patterns with food consumed across the 24 h period, including those at night [5,7]. Shift workers regularly experience circadian misalignment, which occurs when the fast/feeding times are desynchronized with the temporal pattern established by the central circadian clock [8]. In relation to this, the changes in the intake of food especially among shift workers are known to influence many elements of cognitive performance, emotional state, and wakefulness [9]. For example, the cognitive-behavioral consequences of food intake restrictions for the short term are associated with lack of energy supply, whereas the long-term effect involves lack of supply of essential nutrients [10]. The intake of nutrients, thus, plays a decisive role in the regulation of the nervous system and behavior [11].

It is important to highlight the concern of nutritional intake and psychosocial dilemma among shift workers in order to design and promote suitable lifestyle practices for the population. Hence, in the present review, we aimed to explore the impact of circadian disruption in individuals working in shifts and the effect on their chrononutrition and psychosocial well-being.

## 2. Materials and Methods

This was a narrative review. A literature search for relevant articles was conducted in October 2019 using databases such as Google Scholar, PubMed, and Scopus. The studies were identified on the basis of information available in the title and abstract. Relevant search terms included “circadian rhythm”, “biological clock”, “shift work”, “psychological”, “nutrition”, “diet”, “shift work and circadian rhythm”, “shift work and dietary habit”, “shift work and psychological”. We found 110 articles on the basis of the above search approach. All the studies were limited to the publications in the English language, and we took into account all publications from 2000 until 2019.

## 3. Results and Discussion

The results and discussion section outlines the nature of shift work, sleep/wake cycle, chrononutrition, daytime meal, late-night meal, dietary habit and meal changes due to shift work, altered sleep/wake cycle and nutritional intake, psycho-social well-being and nature of work, as well as the relationship of nutritional intake and psychosocial well-being among shift workers.

### 3.1. Nature of Shift Work

There are different working hours according to each country, e.g., approximately 35 h per week in France [12], 37 h per week in Denmark [13], and more than 40 h per week in United States (US). In accord with the US, the Association of Southeast Asian Nations (ASEAN) countries, which include Brunei, Cambodia, Indonesia, Laos, Malaysia, Myanmar, Philippines, Singapore, Thailand, and Vietnam, also have more than 40 h of working hours per week [12,14]. Shift work is characterized by working during daytime hours, which comprises fluctuating or rotating patterns among night, early morning, and evening schedules [15].

The magnitude of shift work is regularly growing in the modernized environment as a result of an ascending financial system and enhanced global transportation [16]. Night shift is characterized by the time of the work interval between midnight and five o’clock in the morning, and the shift lasts for at least seven consecutive hours. Shift work allows continuous operation of certain industries by establishing a rotation work schedule among employees [17]. Shift work schedules vary greatly with regard to timing and duration for every shift, and the speed of shift rotation. However, it is widely accepted that prolonged exposure to a shift work schedule is related to increased health complications compared with those working at normal daytime hours (Table 1) [18]. According to World Health Organization (WHO), the definition of “health” is a state of complete physical, emotional, as well as social wellness, and not only mere absence of illness or ailment [19].

An earlier study by Atkinson et al. (2008) reported numerous adverse effects on health outcomes related to shift work, including increased risk of sleep alteration and fatigue, cardiovascular diseases, and obesity [20]. Physiologically, sleep disturbance among shift workers is a result of desynchronization between light–dark phase, sleepiness, and intake of food [8]. This, in turn, gives impact to their physical and psychological wellness and also negatively affects the performance at the work place [21]. Figure 1 illustrates the effect of circadian rhythm disruption on different body systems that include cardiovascular, gastrointestinal, and central nervous system [22,23,24,25,26,27,28].

### 3.2. Sleep/Wake Cycle (24-h Circadian)

Humans are mainly active or awake during the day and inactive or asleep at night. The cycle is in synchrony with the light and dark cycle during day and night. Nevertheless, this has evolved remarkably since the 19th century with the artificial light discovery that directed to the opening of the first power plant, which consequently led to the availability of a continuous and reliable point of supply for the electrical power through day and night. The advancement of modern technology provides the artificial light to brighten the night. Hence, this condition is capable to alter the humans to be active at night during the normal sleeping time. When the normal synchronization of the sleep/wake and light/dark cycles are disrupted, humans override the circadian control of the sleep/wake cycle, and that, in turn, promotes internal desynchronization between circadian rhythm and sleep [25,29].

The circadian system coordinates physiology and behaviors towards the environment in a way that the body works as a finely harmonized clock. The suprachiasmatic nucleus (SCN) of the hypothalamus acts as the master clock, synchronizing 24 h rhythms in physiological behavior in the body, inclusive of other brain regions and peripheral tissues. When aligned accordingly to the environment, the clock stimulates sleep and related anabolic functions at night, such as immune function and hormone release, and wakefulness and its associated catabolic functions during the day, i.e., food intake and metabolism, physical activity [30,31]. In addition, the endogenous melatonin rhythm that is controlled by the SCN clock often demarcates the internal biological day and night. For instance, in humans, there is a high level of melatonin secretion during the biological night, and low melatonin level occurs during the biological day [23,32]. Moreover, similar clock oscillators to the SCN clock were identified in peripheral tissues such as gastrointestinal tract liver, muscle, or adipose tissue [33]. It has been shown that consuming food which sometimes contradicts our circadian rhythms can entrain rhythms in peripheral tissues, like liver [34], resulting in long-term health issues [35]. Desynchronization caused by feeding or alternative mechanisms of entrainment in a population of shift workers can lead to defective use of substrates, resulting in the disruption of metabolic pathways, thereby leading to intramyocellular accumulation of lipids and insulin resistance [8].

The natural periodic condition of rest for the mind and body is by having adequate sleep, and the average duration of sleep for an adult is 7 to 9 h per day, as per recommendation by The National Sleep Foundation [36,37]. The individuals may act and think slow, and have tendencies to create more errors if they do not get sufficient sleep. Consequently, sleep deprivation can lead to susceptibility of having typical viral infection, diabetes, obesity, heart disorder, as well as depression [38]. This is demonstrated in Figure 1, i.e., the effect of circadian disruption on important systems in the body.

Reduced sleep quality influences significant changes in the eating behavior among shift workers by boosting their appetite late night, thereby leading to obesity [39,40,41,42,43]. While there is evidence that leptin and ghrelin are altered with sleep constraint, thereby raising appetite at night, there is also evidence that factors other than hunger drive shift workers to eat throughout the night shift. For instance, due to the time available and break availability, there is social pressure to eat with colleagues, as a strategy to stay alert, avoiding gastric upset as well as stress eating [44]. It was also emphasized by Waterhouse et al. (2003) that some of the significant factors of shift workers eating at night are due to habits, time pressure, and social factors [6]. Besides, colleagues also act as significant role models, where they copy positive and negative attitudes from one another [45]. The timing and meal quality are limited by job requirements in the workplace and the social, domestic, and rest requirements outside the workplace [5].

For humans, early morning exposure to bright light helps earlier circadian rhythm advancement progress. Conversely, early night exposure to bright light delays these rhythms [46]. The light exposure pattern can be designed to reset the central circadian pacemaker quickly to earlier or later stages [47]. The shift workers’ sleep schedule is erratic and always abruptly displaced at abnormal circadian phases [46]. The majority of the night shift workers have different degrees of circadian adaptation to their schedules of work [48]. Individuals vary widely in their degree of tolerance towards shifting work because of the intensity of circadian and sleep–wake disruptions [49]. Furthermore, it was reported that the circadian pacemaker’s full adaptation to night shift work would only take place in a minority of workers, even if they work on a fixed night shift schedule [50]. Hence, it is difficult for the adaptation to circadian rhythm to happen among shift workers in view of the rotating shift schedule which is not fixed.

In a shift work simulation study, subjects that develop sleep deprivation tend to choose low nutritional value snacks together with sweetened beverages during odd times of the day [39]. Morris et al. (2015, 2016) confirmed that circadian rhythm misalignment stimulates metabolic dysfunction that eventually leads to weight gain [51,52]. The biological clock performs the function to fix to an energy restrain mode after late night. Therefore, intake of energy-dense food at night promotes rapid fat deposition. In relation to this, meal consumption at night also demonstrates impairment of lipid and glucose tolerance in view of hormonal disturbance in the metabolism [28,53,54]. The health impact with shift work is represented in the form of a table (Table 1).

### 3.3. Chrononutrition

Chrononutrition is a term that describes food consumption in consideration of the meal timing. Normally, three elements of time are acknowledged, i.e., (1) inconsistency (irregular routine of eating), (2) frequency (the number of daily meals), and (3) clock time (definite time of intake) [58]. The most obvious timings of intake in discussing eating behavior are the breakfast, lunch, tea, dinner, and supper or late-night eating. Previous studies highlighted that changes in the intake for breakfast and supper have a significant impact on the body weight and health [59].

The gastrointestinal tract is lined with enteroendocrine cells that provide endless supply of different hormones based on cues from ingested foods [60]. Subsequently, signals are sent via gut–brain axis to the brain. Energy balance and homeostasis are achieved only by symphonious response of peripheral metabolism in the body. The favorable process of metabolism demands well-regulated eating habits as hormonal regulation occurs at a specific period of time to ensure the optimal process to take place, not too rapid and not too slow. Interestingly, the entire timely process works efficiently based on specific types of macronutrients [61].

The first meal of the day regulates the determination of peripheral clocks’ circadian rhythm, meanwhile, the last meal of the day brings about the process of lipogenesis and accumulation of adipose tissues [62]. Previous studies pointed out that activation of lipolysis was prolonged and lipogenesis process increased among the breakfast skippers [63,64]. There was evidence that night shift workers skipped their breakfast due to the nature of shift that ended early in the morning. It was also reported that the workers preferred to sleep after the night shift [65]. On the other hand, another study showed that post night shift workers often consume a heavy breakfast, followed by sleep [66]. Another example is that of the shift workers who might eat breakfast at the end of a night shift before driving home, while few might eat at home before their daytime sleep, and also others who eat with their family when they get home after a night shift.

It has been understood that since breakfast usually follows the longest fasting period during the 24 h daily cycle, the omission of breakfast could significantly alter metabolism and disrupt the function of gastrointestinal tract. This subsequently leads to reduced availability of nutrients to the brain and likely adverse behavioral outcomes [67]. The effect is similar in those who work normal working hours, but study has shown that shift workers are more susceptible to breakfast skipping. For instance, a study by Axelsson et al. (2004) found that the shift workers might be inclined to skip breakfast so as to allow more sleeping time [49].

However, another important issue that needs to be emphasized when targeting the time of the first meal of the day is the setting of insulin secretion. As insulin fluctuation is closely dependent on meal timing, eating the first meal of the day too early with an insufficient restraining food intake period may result in a weak resetting effect of insulin secretion. All these mechanisms contribute to significant implications on weight gain, appetite, as well as glucose and lipid metabolism [35,62,68]. Morris et al. (2016) also proposed that irrespective of the behavioral cycle consequences, glucose tolerance was lower in the biological evening than in the morning. Thus, it indicated that one of the important factors to consider in shift workers is the internal circadian time of food intake [52].

### 3.4. Daytime Meal

Eating breakfast is considered to have good effect on the quality of diet throughout the day [69]. Specifically, breakfast is being described as a fundamental aspect in daily nutritional necessity, which is also partly responsible for nutritional quality and energy intake [70]. The definition of breakfast is the earliest eating event that takes place in a day, within the timing of two hours of waking up from sleep, and approximately before 10:00 am [71]. Based on previous studies, it is reported that diseases of dyslipidemia [72], hypertension [73], diabetes mellitus [72], coronary heart disease [74], and weight gain [75] are associated with skipping of breakfast. However, the timing of breakfast among shift workers is subjected to the timing of last energy intake on the previous day, in order to ensure a sufficient fasting period for the body to reset its metabolism.

Barr et al. (2013) examined the association between breakfast, nutritional intake, and nutrient adequacy among Canadian adults in order to specify the type of nutrients to be consumed. Breakfast consumers, especially ready-to-eat cereal breakfast consumers, had positively higher fiber intake as well as several vitamins and minerals compared with breakfast skippers [76]. Besides, a survey of the Bogalusa Heart Study (*n* = 504; 58% women; 70% white; age of 19 to 28 years old) reported 74% of breakfast skippers did not achieve two-thirds of the Recommended Dietary Allowance for vitamins and minerals, compared with 41% of the breakfast consumers [77]. Furthermore, consumption of protein during breakfast causes better initial and sustaining feeling of fullness, greater satiety, and low concentration of the appetite-regulating hormone ghrelin [78,79,80]. Almoosawi et al. (2013) recommended that a long-term protective effect against metabolic syndrome can be obtained by increasing the intake of carbohydrate in the morning [81].

### 3.5. Late-Night Meal

Shift workers were found to have nibbling behavior during their night shifts, mainly high-carbohydrate food. The nocturnal feeding imitates night eating syndrome (NES) effects that were first described by Stunkard et al. in 1955 [82]. Multiple studies by St-Onge et al. (2017), Berg et al. (2009), and Cleator et al. (2012) reported that late-night meal was associated with obesity and cardio-metabolic health [83,84,85]. There were no reports on associated similar psychological outcome among patients suffering from NES and shift workers who had late-night eating habits.

Compared with high-fat diets, consumption of foods rich in carbohydrate at night creates a greater increment of sleepiness level and a decrement in mental performance, contrary to the physical act [86,87]. Meanwhile, there was an association between protein intake and diminished feeling of hunger, increased satiety, and reduced caloric intake, in comparison with other macronutrient intakes [88]. Protein-rich meals promote higher satiety and alertness due to the thermogenic effect [89]. Thus, it is recommended for the shift workers to consume high-protein diet at night and reduce the intake of food that is rich in carbohydrate in order to have a productive night shift work.

### 3.6. Dietary Habit and Meal Changes due to Shift Works

According to the systematic review and meta-analyses by Bonham et al. (2016), it was reported that the energy intake of shift workers lasting 24 h did not vary from that of fixed day workers [90]. It was suggested that eating meals specifically at the wrong time of the 24 h cycle was a key contributor to the increased risk of metabolic disturbances in shift workers [91]. Hence, it was obviously shown that meal timing and chrononutrition was important to consider.

A study conducted by Gifkins et al. (2018) among nurses working in shifts found that there was increased food craving, caffeine consumption, and snacking behaviors throughout the night shifts as well as the inability to consume enough fluids at work. The experienced nurses described more about meal skipping at work and relation with a tremendous workload as well as consumption of alcohol as the approach to rest from shift work [27]. Likewise, it was reported among the mine workers who worked in rotating shift duties that they had difficulty in following typical patterns to consume meals [92]. Meanwhile, for other jobs like flight attendants and long-distance truck drivers, the choices about the meal timing and availability of food were restricted in view of fluctuating work schedules [93,94].

Shift work was also strongly related to abnormal eating habits among nurses according to a study done by Wong et al. (2010) in a major acute hospital in Hong Kong. The Dutch Eating Behavior Questionnaire (DEBQ) scores from the participants for components of abnormal emotional, external, and restraint were 66.4%, 61.4%, and 64.0%, respectively. The results also showed that the nurses having four or more shift works in a month were more likely to demonstrate DEBQ scores of abnormal emotional (adjusted odds ratio (aOR) 2.91%, 95% C.I 1.57–5.42, *p* = 0.001) and restraint (aOR 3.35, 95% C.I 1.76–6.38, *p* < 0.001) [95]. Another study on chrononutrition emphasized that those who were having more erratic meal routine had higher risk of developing obesity and metabolic syndrome, despite using less energy than those who had regular meal patterns [58]. Other studies showed lower quality of diet in shift workers, such as greater consumption of sugar [96], saturated fat [97,98], and sweetened beverages [96,99], and poorer intake of vegetables [98,100]. Moreover, hyperphagia, obesity, and insulin resistance were seen in view of circadian rhythm misalignment [101,102,103]. This was because of the regulation of circadian expression and activity of enzymes and hormones engaged in metabolism, which was altered due to the disruption in the clock regulation in the brain and peripheral tissues [26,104,105].

Strickland et al. (2015) reported that the duration of time breaks during working late shifts has a significant effect on the healthy eating among shift workers. Time restriction impedes consumption of healthy food, reduces satiety, and increases binge eating [106]. The prevalence of eating disorder was also found to be higher among shift workers with extreme job stress, reflecting that there was inter-related relationship between psychosocial well-being and nutritional intake [107]. Similarly, an earlier study demonstrated an unusual pattern of restrained or binge eating among shift workers who struggled with high stress [108].

Researchers also suggested that working in shift schedules alters not only the usual meal timing, but also the accessibility of healthy meals, hence, giving rise to multiple health-related problems [96,109]. Shift workers who work during night shifts are likely to consume fast food and processed food with high content of sugar or salt from prepackaged meals or vending machines, due to less availability of food services [6]. Night shift workers frequently experience decreased stamina and sleepiness at work, and while this may be linked to the high content of sugar and the availability of food, it is largely due to the circadian influence of sleep pressure that is greatest at night [110].

Furthermore, there was a considerably larger proportion of energy from sugar consumed over a 24 h period with night shift workers compared with day shift workers [111]. De Assis et al. (2003) found that there was no significant difference in total food consumption between morning, afternoon, and night shift workers. Nevertheless, it was reported that morning shift workers consumed more energy and macronutrients in the morning than the other shift workers. In comparison to the morning shift workers, the afternoon and night shift workers had a higher energy consumption at noon and at dawn, respectively. Meanwhile, throughout their work shifts, night shift workers and morning shift workers ingested high-energy foods and drinks, compared with afternoon shift workers. It can be established that while total energy consumption and composition are less influenced by changes in shift work over a 24 h span, the frequency of meals was reduced and the prevalence of high-energy snacking during the specific shift worked was increased [96].

Figure 2 illustrates the regulation of hormones that synchronically regulate metabolism based on the biological clock [112].

Chronobiologically, late-night meals and sleeping during the day disrupt the regulation system and affect the individual’s appetite as well as metabolism. Furthermore, during daytime, humans are predisposed to the promotion of glucose metabolism and fat storage when they normally eat, meanwhile, at night, they are predisposed to glucose sparing and fat metabolism when they normally fast. Due to this predisposition, shift workers present a lowered glucose and lipid tolerance consecutively because of the change from day to night duties [113].

A study by Bandin et al. (2015) examined the influence of meal timing changes on the energy expenditure, glucose tolerance, and circadian-related variables, in which the participants were provided with standardized meals during the two meal intervention weeks and were observed under two lunch eating conditions: early eating (lunch at 13:00) and late eating (lunch at 16:30). It was found that eating late was associated with decreased resting energy expenditure, decreased fasting carbohydrate oxidation, decreased glucose tolerance, blunted daily profile in free cortisol concentrations, and decreased thermal effect on food. It also emphasized the implications on the differential effects of meal timing on metabolic health in this study [54].

### 3.7. Altered Sleep/Wake Cycle and Nutritional Intake

Baron et al. (2011) and Hsieh et al. (2011) identified that there was an association between short sleep duration of less than 5 h or late sleepers (midpoint of sleep > 5:30 am) and eating late-night meals or consuming more calories late in the evening with significantly greater risk for having obesity and diabetes [53,114]. In addition to that, a study by Colles et al. (2007) also demonstrated that night eating syndrome characterized by a time-delayed eating pattern was strongly related to increased body mass index (BMI) [115].

Meal timing has an important impact on circadian rhythm in peripheral organs, causing feeding time and circadian clocks to be tightly intertwined [116]. Feeding–fasting cycles are generated by time-restricted feeding which consolidate circadian rhythmicity in gene expression and numerous metabolic pathways’ circadian activation. This is because the clock in a majority of the peripheral organs readily responds to feeding cycles, and feeding time can shift their period. Upon a few days of time-restricted feeding, the availability of food and the endogenous clocks are coordinated, irrespectively of whether the meal is served in the light or dark phase. Thus, diet which contains high fat alters circadian rhythmicity through dampening of feeding–fasting cycles [117].

### 3.8. Psychosocial Well-Being and Nature of Work

Working conditions play an increasingly authoritative role in psychological and mental well-being among changing social circumstances. Due to its nature, psychosocial work conditions could not be characterized by direct measurements in comparison with physical or chemical hazards [118]. Psychosocial factors include aspects like social support in the workplace, job satisfaction, or physical workload [119]. Another study by Goetz et al. (2015) found that there were a variety of psychosocial factors which correlated with burnout, such as symptoms of higher cognitive stress, conflicts between job and privacy, emotional demands and role-conflict, lower general health, satisfaction with life, demands to conceal emotions, and younger practical assistants [120].

Shift workers involved in night shifts reported more psychological and mental health issues than day workers. The problems comprise irritability, somatization, obsessive-compulsive disorder, interpersonal sensitivity, anxiety, altered mood, and paranoid disorders that had more prevalence predominantly [121]. Besides, other studies also showed the average sleep quality among night shift workers to be relatively lower compared with day and no night rotating workers [122,123]. These data highlight that working in shifts especially in night schedule has negative effects on psychological and social well-being [124].

### 3.9. Relationship of Nutritional Intake and Psychosocial Well-Being among Shift Workers

Based on a recent scoping review by Gupta et al. (2019), the emotional state resulting from a night shift is reported to have an effect on food consumption, not just on-shift but on the day after [44]. Shift workers may cope with the stress associated with work by consuming more food than usual, such as increased consumption of junk foods [125,126]. Junk foods are known as energy-, fat-, sugar-, and/or sodium-dense foods [127], but deprived of vitamins and micronutrients [128]. These are low-quality food choices, also called calorie-dense food, which is nutritionally poor and contains many more calories than nutrients like vitamins and minerals [129]. Earlier studies suggested that with the contrast of high-fat with high-carbohydrate meals, major postprandial effects of diet, or interactions with time of day, a variety of mood states, including alertness, boredom and mental slowing, were observed [130].

Stress eating is a common phenomenon [131], and shift work is linked to high rates of work stress [132]. In this sense, eating on shift, while bad for long-term health, may be important for coping on shift. Shift workers usually report having irritability, nervousness, and anxiety in association with stressful working conditions as well as hardships in family and social life. Continuous circadian rhythm disruption and sleep deprivation may complicate with chronic fatigue, mood disorders, neuroticism, chronic anxiety, or depression, subsequently leading the workers to have higher absenteeism and frequently need the administration of psychotropic drugs like sedatives and hypnotics [133].

Moreover, there is a syndrome known as “shift work disorder” that has been proposed recently, and it is characterized by the presence of the following symptoms: disrupted circadian rhythm of sleep/wake, insomnia, fatigue, and excessive day sleepiness [134,135]. Daily levels of alertness and job performance are altered by the night shift duties as they induce sleep deprivation [136,137]. There are also negative impacts on the quality of life among the shift workers, as seen with greater divorce and suicidal rates, poor morale, higher usage of alcohol and drugs, and a sense of malaise provoked by the deficiency of pleasure in the domestic as well as social areas of life [138].

Ferri et al. (2016) highlighted that nurses with rotating night duties require specific consideration because of greater risk for job dissatisfaction and negative health outcomes. This cross-sectional study investigated the comparison between nurses with night shifts and nurses with day shifts, with the association of risk factors predisposing nurses to worse health effects and lower job satisfaction. The results showed a statistically significant association in the nurses engaged with rotating night duties, where they reported the lowest mean score in the items of job satisfaction, quality and quantity of sleep, with more frequent chronic fatigue, psychological and cardiovascular symptoms as compared with the day shift workers [24]. Depressive symptoms were more frequently related to night eating syndrome [139,140,141]. However, it may be affected by the concurrence of binge eating disorder, which is highly associated with psychological distress [142]. In addition, there is a link between binge eating disorder and night eating syndrome [143,144] as well as nocturnal snacking [145,146] in obese population, but the description of the relationship still remains unclear.

Increased intake of snacks and fast food among shift workers results in high intake of energy-dense and high-fat diet. In an earlier study conducted by researchers from Arizona university, it was found that 60% of individuals admitted to night-time snacking and lack of adequate sleep, which led to craving for more junk food [147]. The researchers held the view that inadequate sleep, junk food consumption, and unhealthy night-time snacking represented the fact that sleep helps regulate metabolism [147]. Sleep may give the individual a better feeling of overcoming stress and somehow protect from unhealthy eating. When food is taken late at night, the body may not be able to burn enough calories, and they may be stored. Unless there is an activity at night, all consumed food may cause harmful effects. Sedentary jobs such as working at any call center or any office where movement is minimal mean there is more likelihood of food being digested at a different rate. Diurnal and nocturnal variation is also hormone- and temperature-related, and it may influence the metabolism in any individual.

The fatigue in shift workers may explain the reluctance for them to prepare healthy meals, alternatively substituting with industrial snacks, fast food, and sweets [148]. Table 2 describes the prevalence of overweight and obesity among shift workers. Hence, the prevalence indicates that there is an urgency to have nutritionists/dietitians in the 24 h industries that specifically look into the well-being of the shift workers on their nutritional status.

## 4. Conclusions

In a nutshell, the nutritional intake and psychosocial well-being in shift workers need further attention. This is due to the occurrence of circadian rhythm misalignment when the individual works in the shift duties, subsequently causing the meal timing to be interrupted. However, it is not relevant to get rid of the shift work system in society in view of the expanding twenty-four-hour industry, globally. Shift workers need to assure the operation of 24 h basis to take place smoothly. Thus, proper meal intervention for the shift workers has to be emphasized in terms of the type of food to be consumed during shift duties, the timing of meal to be taken, as well as how to have a healthy diet. Further studies are recommended to promote good nutritional intake and psychosocial well-being among shift workers in our society.

## Figures and Tables

**Figure 1 ijerph-17-02043-f001:**
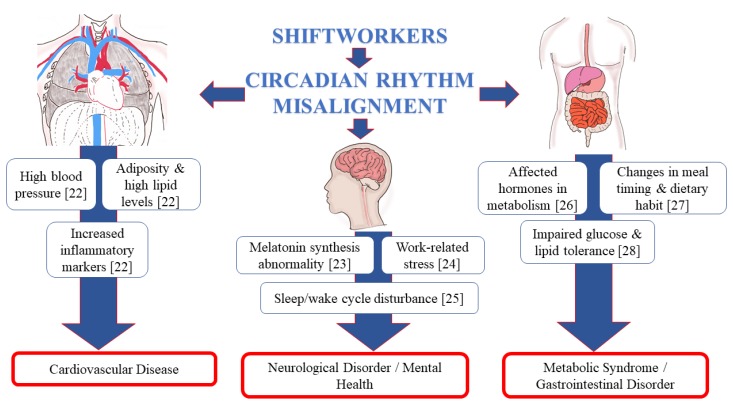
Schematic diagram on the effect of circadian rhythm disruption on different body systems.

**Figure 2 ijerph-17-02043-f002:**
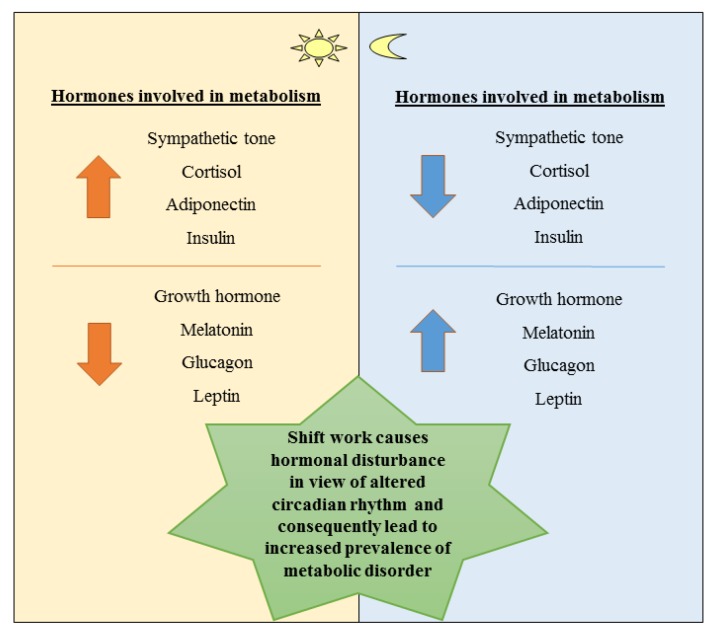
Schematic diagram showing effect of different hormones based on circadian rhythm. Adapted from Bass et al. (2010) [112].

**Table 1 ijerph-17-02043-t001:** Impact of shift work on health of individuals.

Study & Country	Method	Sample Size (*n*)	Disease Related to Shift Works	Description on the Disease
Nikpour et al. [55] (Iran)	Cross-sectional study	209	Metabolic syndrome	Diagnostic criteria: simultaneously met three out of five criteria: (1) Hypertension, BP > 130/85 mmHg; (2) High serum TG level > 150 mg/dl; (3) High FBS > 110 mg/dl; (4) Low serum HDL level < 50 mg/dl; (5) Abdominal obesity, WC > 88 cm
Thomas et al. [22] (United Kingdom)	Large population-based cohort study	7839	Cardiovascular disease	Risk factors of cardiovascular disease such as adiposity, blood pressure, blood lipids, blood glucose, and level of inflammatory factors
Koh et al. [56] (Korea)	Cross-sectional study	203	Gastrointestinal disorders	Diseases include irritable bowel syndrome, functional dyspepsia
Knutsson et al. [57] (Sweden)	Longitudinal cohort study	549	Breast cancer	Increased risk for breast cancer among women who work in night shifts
Ferri et al. [24] (Italy)	Cross-sectional study	213	Psychological disorders	Job dissatisfaction, poor sleep quantity and quality, chronic fatigue, psychological stress

Note: BP, blood pressure; TG, triglyceride; FBS, fasting blood sugar; HDL, high-density lipoprotein; WC, waist circumference.

**Table 2 ijerph-17-02043-t002:** Prevalence of overweight and obesity in workers from different parts of the world.

Study & Country	Sampling Frame	Method	Sample Size (*n*)	Criteria Used	Prevalence of Overweight/Obesity
Zhao et al. [149] (Australia)	Nurses and midwives	Cross-sectional study	1235	WHO	32.9% overweight; 27.4% obesity
Canuto et al. [150] (Brazil)	Shift workers in poultry-processing plant	Cross-sectional study	580	WHO	11.2% obesity
Kubo et al. [151] (Japan)	Shift workers manufacturing industry-based corporation	Retrospective cohort study	920	Obesity (BMI ≥ 25.0 kg/m^2^)	21.1% obesity
Whitfield et al. [152] (United States)	Long-haul truck drivers	Cross-sectional study	92	WHO	86% overweight; 66% obesity
Guo et al. [153] (China)	Shift workers in motor corporation	Cross-sectional study with retrospective assessment	9088	Obesity (BMI ≥ 28.0 kg/m^2^)	13.5% obesity

Note: WHO, World Health Organization with BMI (body mass index) cut-offs for underweight (BMI < 18.5 kg/m^2^), normal (BMI 18.5 to 24.9 kg/m^2^), overweight (BMI 25.0 to 29.9 kg/m^2^), and obesity (BMI ≥ 30.0 kg/m^2^).

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
