# Peer review of "Consequences of Circadian Disruption in Shift Workers on Chrononutrition and their Psychosocial Well-Being"

_ijerph, 2020, doi:10.3390/ijerph17062043_

Round 1
Reviewer 1 Report
Thankyou for addressing my comments, I am happy with the revised manuscript and believe it to be a good addition to the chrononutrition field.
Author Response
Response to Reviewer 1 Comments
Point 1: Thank you for addressing my comments, I am happy with the revised manuscript and believe it to be a good addition to the chrononutrition field.
Response 1: We thank the reviewer for the comments.
Reviewer 2 Report
The paper is greatly improved, and I thank the authors for considering my comments. I still am struggling with understanding the justification for this paper. You are linking shiftwork to chrononutrition and shiftwork to psychosocial well-being. Yes, shiftwork can lead to issues in both, but why are these two both considered in this one review? Shiftwork leads to many health concerns and poor behaviors. If you linked chrononutrition to psychosocial well-being then it would make sense, but I do not understand why you decided to choose these two aspects of shiftwork. It is almost like you have two mini reviews that you combined into a single review. Frankly, I just do not see the connection being made and wonder why these are two separate papers.
I read through section 3.8. I do not think you are making a strong enough justification to include both chrononutrition and psychosocial well-being. Have you considered just making this a paper on shiftwork and chrononutrition? You do not talk about or mention psychosocial well-being much at all, but yet it is in the title of the paper. If chronutrition has its own sections, why does psychosocial well-being in shiftwork not have its own section? Three things would help: 1) have a section on psychosocial well-being and shiftwork, 2) include a more thorough link between chrononutrition and psychosocial well-being in section 3.8 which you can then use to hypothesis relationships between all three, and 3) create a better justification in the Introduction as to why this paper includes both chrononutrition and psychosocial well-being.
The new text you added in 3.3 doesn’t quite make sense to me. You indicate when some shift workers may or may not eat breakfast, but how does that relate to health in this context. Also, you are equating meals at 6:00AM to 8:30AM as breakfast for shift workers. That isn’t breakfast for them, that is dinner for them. Most of the section on chrononutrition is coming from the perspective of someone working a daytime schedule. This needs to be more in context with shiftworkers. Also, a much more thorough explanation of when shiftworkers actually eat needs to be discussed. There are more than enough studies that discuss the energy distribution on a night shift. That needs to be incorporated.
What does breakfast skipping have to do with chrononutrition in shiftworkers? I understand that breakfast skipping can affect peripheral circadian rhythms, but a shiftworker not eating breakfast between 6AM and 8:30AM as you noted in section 3.3 isn’t the same as skipping breakfast for a someone living by a daytime schedule, so how is this section pertinent?
Lastly, there is no discussion about phase advancement and phase delay. Just because a shiftworker is eating at night doesn’t mean there is anything wrong with their circadian rhythms. If they completely adopt a nighttime schedule every day, they can completely invert their rhythms. There are studies showing it is possible to invert rhythms to be the opposite of a person living a daytime schedule. Therefore, eating at night may not be adversely affecting them. So, I think it is important to talk about rotating shifts and reverting back to a daytime schedule on days off which many shiftworkers do.
Author Response
Response to Reviewer 2 Comments
Point 1: The paper is greatly improved, and I thank the authors for considering my comments. I still am struggling with understanding the justification for this paper. You are linking shift work to chrononutrition and shift work to psychosocial well-being. Yes, shift work can lead to issues in both, but why are these two both considered in this one review? Shift work leads to many health concerns and poor behaviours. If you linked chrononutrition to psychosocial well-being then it would make sense, but I do not understand why you decided to choose these two aspects of shift work. It is almost like you have two mini reviews that you combined into a single review. Frankly, I just do not see the connection being made and wonder why these are two separate papers.
Response 1: We thank the reviewer for the comments. We have added further clarification in the part of introduction discussing about the link between shift work and chrononutrition, as well as shift work and psycho-social well-being. We took into account both the aspects of shift work because we were interested to approach the individuals holistically, in terms of physical health and also psychological health. The added paragraph in line number 45 emphasises that chrononutrition could influence the psychological well-being, including the cognitive performance, emotional state and wakefulness. The cognitive-behavioral consequences of food intake restrictions can be observed in short-terms and long-terms among shift workers.
Point 2: I read through section 3.8. I do not think you are making a strong enough justification to include both chrononutrition and psychosocial well-being. Have you considered just making this a paper on shift work and chrononutrition? You do not talk about or mention psychosocial well-being much at all, but yet it is in the title of the paper. If chrononutrition has its own sections, why does psychosocial well-being in shift work not have its own section? Three things would help: 1) have a section on psychosocial well-being and shift work, 2) include a more thorough link between chrononutrition and psychosocial well-being in section 3.8 which you can then use to hypothesis relationships between all three, and 3) create a better justification in the Introduction as to why this paper includes both chrononutrition and psychosocial well-being.
Response 2: Additional paragraph has been added in new section of 3.8 to describe the psycho-social well-being and nature of work. Furthermore, it has been described in section 3.9 in line 342 as we found that chrononutrition has effect on psycho-social well-being. We also included further justification in the section of introduction (line 45) regarding the fact why we focus on chrononutrition and psycho-social well-being.
Point 3: The new text you added in 3.3 doesn’t quite make sense to me. You indicate when some shift workers may or may not eat breakfast, but how does that relate to health in this context. Also, you are equating meals at 6:00AM to 8:30AM as breakfast for shift workers. That isn’t breakfast for them, that is dinner for them. Most of the section on chrononutrition is coming from the perspective of someone working a daytime schedule. This needs to be more in context with shift workers. Also, a much more thorough explanation of when shift workers actually eat needs to be discussed. There are more than enough studies that discuss the energy distribution on a night shift. That needs to be incorporated.
Response 3: We have elaborated further regarding chrononutrition in context with shift workers (refer to line 178). We have deleted the part of equating meals at 6:00AM to 8:30AM as breakfast for shift workers. Besides, additional paragraph has been added (refer to line 277) about energy distribution among shift workers. We have included the recommendation for the shift workers to consume high-protein diet at night to promote higher satiety and alertness in order to have productive night shift work (line 232).
Point 4: What does breakfast skipping have to do with chrononutrition in shift workers? I understand that breakfast skipping can affect peripheral circadian rhythms, but a shift worker not eating breakfast between 6AM and 8:30AM as you noted in section 3.3 isn’t the same as skipping breakfast for a someone living by a daytime schedule, so how is this section pertinent?
Response 4: We have related between the association of breakfast skipping and shift workers (line 184). Studies have shown that shift workers are more susceptible to skip breakfast. The effect is similar with individuals with normal working hours, in which both could significantly alter metabolism and disrupt the gastrointestinal function.
Point 5: Lastly, there is no discussion about phase advancement and phase delay. Just because a shift worker is eating at night doesn’t mean there is anything wrong with their circadian rhythms. If they completely adopt a night time schedule every day, they can completely invert their rhythms. There are studies showing it is possible to invert rhythms to be the opposite of a person living a daytime schedule. Therefore, eating at night may not be adversely affecting them. So, I think it is important to talk about rotating shifts and reverting back to a daytime schedule on days off which many shift workers do.
Response 5: Additional paragraph has been added (line 137) discussing about the phase advancement and the phase delay among shift workers. The adaptation to circadian rhythm is difficult to happen among shift workers in view of the rotating shift schedule which are not fixed. The night time schedule is not fixed every day, thus it is difficult for the workers to completely adopt and completely invert their circadian rhythms. Besides, majority of the night shift workers have different degrees of circadian adaptation to the work schedule.

Round 2
Reviewer 2 Report
Comment 1: I am glad to see that you added some information related to food nutrition and psychosocial health among shift workers. In looking at that section, I have now realized that one other piece of information would greatly help. I think more text on how certain types of diets (e.g., high fat/high sugar) seen among shift workers are associated with psychosocial health would help. For example, there should be a fair bit of literature relating poor dietary habits to cognition and psychosocial health among anybody. I think some of that needs to be mentioned to really connect dietary changes in shift workers to psychosocial health. You focused your new section on restriction of food, but what about the change in quality of food among shift workers? Granted your paper is focused on chrononutrition not overall quality, but chrononutrition also involves distribution of macronutrients. So, what do studies show of high fat/high carb diets at night on psychosocial health? A couple of sentences of that would be helpful.
Comment 2: The sentence on lines 45-46 needs its own reference.
Comment 3: the new text on lines 99-100 does not grammatically make sense to me.
Comment 4: Not sure the work “served” on line 331 is the correct word. It doesn’t make sense there.
Author Response
Response to Reviewer 2 Comments
Point 1: I am glad to see that you added some information related to food nutrition and psychosocial health among shift workers. In looking at that section, I have now realized that one other piece of information would greatly help. I think more text on how certain types of diets (e.g., high fat/high sugar) seen among shift workers are associated with psychosocial health would help. For example, there should be a fair bit of literature relating poor dietary habits to cognition and psychosocial health among anybody. I think some of that needs to be mentioned to really connect dietary changes in shift workers to psychosocial health. You focused your new section on restriction of food, but what about the change in quality of food among shift workers? Granted your paper is focused on chrononutrition not overall quality, but chrononutrition also involves distribution of macronutrients. So, what do studies show of high fat/high carb diets at night on psychosocial health? A couple of sentences of that would be helpful.
Response 1: We thank the reviewer for the comments. We have highlighted all the sentences emphasizing about the association between nutritional intake and psycho-social health among shift workers. We have added text on the change in quality of food among shift workers in line 345. In addition, we have included the meal quality limitation among shift workers in line 136.
Point 2: The sentence on lines 45-46 needs its own reference.
Response 2: We have added suitable reference for the suggested sentence (refer line 45-46).
Point 3: The new text on lines 99-100 does not grammatically make sense to me.
Response 3: We have edited the text (lines 99-101).
Point 4: Not sure the work “served” on line 331 is the correct word. It doesn’t make sense there.
Response 4: We have rephrased it with “involved in” (line 332).

This manuscript is a resubmission of an earlier submission. The following is a list of the peer review reports and author responses from that submission.
Round 1
Reviewer 1 Report
The authors have written a narrative review discussing the relationship between nutritional intake and psychosocial wellbeing of shiftworkers. This was a good review to read and I agree that chrononutrition is a very important concept to explore for the long-term health of shiftworkers. This is a succinct and interesting review that introduces shiftwork and circadian rhythms, as well as the concept of chrononutrition. I have made a number of suggestions below, that may increase the readability of your review. Additionally, there are a number of spelling and grammatical errors throughout the manuscript.
Introduction:
Some examples of different shiftworker industries may be beneficial in the introduction to provide some context for the reader Consider providing some more explanation on the relationship between shiftwork and nutritional intake. For example, mentioning that shiftworkers who work around the clock are likely to eat around the clock, and this is in direct contradiction to circadian rhythms. Emphasising this point may help to preface the explanation about the effects of food intake on the circadian system.Method:
Although this is a narrative review, more detail on the search would benefit your methods section and improve the reliability of results. Details on what databases were searched, and what the inclusion criteria were would be useful to add.Results:
Given the systematic nature of the search, consider adding a small paragraph at the start of the results section that outlines how many articles were included in this review and how the results section is structured (for example, the results section will outline the nature of shiftwork, then will discuss…) Line 75 should read “eventually leads to weight gain” Line 71-80: In my opinion discussing eating behaviours here seems to disrupt the flow of your argument. I suggest moving this to the end to the end of section 3.2 to link to the chrononutrition paragraph. In this paragraph I would also consider including some information about 24h rhythms in metabolism and digestion that are interrupted with shiftwork and people eating at night. Line 72: Although there is evidence that leptin and ghrelin are altered with sleep restriction and therefore appetite is increased at night, there is evidence that reasons other than appetite motivate shiftworkers to eat during the nightshift. For example, social reasons, to stay awake, family reasons, because there is break time available. Consider including some of these reasons in this argument to provide a holistic picture of why shiftworkers eat at night (see work by Lowden et al., Persson et al., Waterhouse et al., Gupta et al.,) Paragraph starting on line 94: Also consider discussing that food timing can entrain peripheral clocks, e.g. in the liver, and this is linked to long-term health issues Line 144 : consider that some nightshift workers do eat breakfast as part of their 24h eating patterns, for example those that eat breakfast at the end of a nightshift before driving home, those that eat at home before their daytime sleep, those that eat with their family when they get home from a nightshift etc. Section 3.5: I’m not sure the relevance of night eating syndrome in your review about shfitworkers. Consider removing this, or including some more evidence to link this to shiftworkers. Section 3.6: There is evidence from a recent meta-analysis by Bonham et al. 2019 that it is not the amount of food that differs between dayshift and nightshift, but the timing of food intake. Including this information may strengthen your argument that meal timing and chrononutrition is important to consider Line 210: Nightshift workers do frequently experience decreased stamina and sleepiness at work, and while this may be linked to the high content of sugar and the availability of food, it is largely due to the circadian influence of sleep pressure that is greatest at night. Section 3.8: it could also be mentioned in this section that shiftworkers report eating to decrease stress levels on-shift. This has been reported in a recent literature review by Gupta et al., 2019. In this sense, eating on shift, while bad for long-term health, may be important for coping on shift.
Reviewer 2 Report
Reviewer comments for ijerph-710425
Overall: This is a review paper that discussed the effect of circadian disruption experienced by shift workers on chrono-nutrition and psychosocial health. This review showed that there are detriments in meal timing and psychosocial health experienced by shift workers which is a growing concern, especially given the focus and attention being paid to chrononutrition in the last decade or so. One comment that should be addressed throughout is the English grammar and scientific writing. I strongly suggest a careful review for these. The text was readable, but there were numerous grammatical errors which did detract some from the importance of this work. I did not point out these grammatical errors in my comments below but would be happy to provide specific edits should this be requested.
Abstract:
On line 16, make sure you mention that night work hours lead to circadian dysregulation. As it is written now, any kind of work hours (including normal day hours) could lead to issues, which is not correct. I think you can combine the statements on lines 17-19 as they mostly relate to the same thing. That will give you more space in the abstract to add something else. What changes are you referring to on line 21? The little bit of text on line 24 doesn’t quite fit with the rest of that sentence and sounds out of place.
Introduction:
Not sure what you mean by the sentence on line 31. It isn’t specific enough. What do you mean by “work regularly based on shift schedules”? I understand this is a review paper and you will discuss some of these topics in the Results section, but there are several sections strongly lacking in the Introduction. You need to justify the need for this systematic review paper. You discuss the problem to some degree, but make no links indicating the possible relationships between shiftwork, chrono-nutrition, and psychosocial health. This needs to be improved considerably.
Methods
The methods also need considerable improvement. Here is a list of things that need to be improved/included: What specific search terms did you use? You have search term listed, but not how you searched them. Did you use AND or OR between the words for example? Also, put quotation marks around each term so people know how you actually used those terms. What databases did you use to find papers? Did you review references of the included papers to look for others? Also, you need to mention what MeSH terms you used. Does “shiftwork workers” pick up “shift work”, “night work”, “irregular hours”……etc.
Results
What are ASEAN countries? I read through each section in the results, but I am not provided specific comments on each section. There is one overall issue that is preventing interpretability of this report. Sections 3.1-3.5 are related to more general background. They do not directly address the problem posed by the title or the aim of the study. Those are more supporting background or information to support why there needs to be a review on shiftwork, chrono-nutrition, and psychosocial health. I feel like these sections are hitting several random topics that are not linked very well together. Sections 3.6-3.8 are more inline with what I would expect the Results to be for this review. Also, your tables do not show specific studies related to chrononutrition in shift workers or psychosocial health and shift workers. The tables show more general topics that have already been reviewed in detail. In short, there needs to be a significant amount of reorganization, more clarity as to what the review is focusing on, and better flow between sections.